# Using Wearable Sensors and a Convolutional Neural Network for Catch Detection in American Football

**DOI:** 10.3390/s20236722

**Published:** 2020-11-24

**Authors:** Bernhard Hollaus, Sebastian Stabinger, Andreas Mehrle, Christian Raschner

**Affiliations:** 1Department of Mechatronics, MCI, Maximilianstraße 2, 6020 Innsbruck, Austria; sebastian@stabinger.name (S.S.); andreas.mehrle@mci.edu (A.M.); 2Department of Sport Science, University of Innsbruck, Fürstenweg 185, 6020 Innsbruck, Austria; Christian.Raschner@uibk.ac.at

**Keywords:** sensor platform, wearable, machine learning, convolutional neural network, American football, catch detection

## Abstract

Highly efficient training is a must in professional sports. Presently, this means doing exercises in high number and quality with some sort of data logging. In American football many things are logged, but there is no wearable sensor that logs a catch or a drop. Therefore, the goal of this paper was to develop and verify a sensor that is able to do exactly that. In a first step a sensor platform was used to gather nine degrees of freedom motion and audio data of both hands in 759 attempts to catch a pass. After preprocessing, the gathered data was used to train a neural network to classify all attempts, resulting in a classification accuracy of 93%. Additionally, the significance of each sensor signal was analysed. It turned out that the network relies most on acceleration and magnetometer data, neglecting most of the audio and gyroscope data. Besides the results, the paper introduces a new type of dataset and the possibility of autonomous training in American football to the research community.

## 1. Introduction

Data in sports is becoming increasingly important. In sports such as football, tennis, baseball, etc., data is used to prepare for matches, improve the performance of athletes and teams, analyze opponents, monitor training progress and many more things [1,2,3,4]. Decades ago, all mentioned tasks were based on human analysis, evaluation and intuition. As possibilities increased with the introduction of computers, computer aided analysis became increasingly popular, e.g., in scouting, opposition analysis or training monitoring [2,5,6,7]. Meanwhile, the state of the art comprises a mixture of human and computer-aided evaluation and analysis, mainly based on data, processed with modern methods of deep learning [8,9,10,11,12].

All computer-based forms of evaluation and analysis need underlying sensor systems to gather relevant data. In American football, these sensor systems are almost exclusively electronic performance and tracking systems (EPTS), according to the definition by the digi sporting consortium [13]. The sensor systems used to record data are dictated by the application and the required data quality, which has lead to a wide range of sensor systems.

First and foremost, optical-based tracking systems and local positioning systems are used across a wide range of sports [4,14,15,16] to determine an athlete’s position on the field and its change over time. For American football in particular, these systems may be combined with complimentary systems, such as wearables for motion sensing [17] or passing machines [18]. The former often come in the shape of inertial measurement units (IMUs). Since these IMUs gather data for a single point in space the placement of the IMU is critical for the measurement [17,19,20,21,22]. Depending on the application the IMU may be placed directly on the skin [17] a wristband, harness or similar [23,24], but also within a ball, shoe or alike is common [25]. If the orientation of the IMU sensor in space is relevant as well, magnetic flux densities are often measured as well in combination with applying some sort of sensor fusion (Kalman-, Madwick-filtering or other [26,27]) on the measured data.

Once data is collected with the aforementioned measurement systems, it is processed with specific algorithms starting from simple Butterworth filters to more advanced sensor fusion techniques [28,29,30]. Over the last decade, processing the given data with neural networks became more popular [11,31,32]. Machine learning provided new analysis methods for data, hence old data could be used to gain new insights [10,32,33,34]. The main problem with machine learning at the moment is the amount of data is necessary to get good generalized results. In some sports, data is practically non-existent, due to the lack of sensors or adequate equipment. Sometimes, data exists but only in a very limited fashion. As a result, applying machine learning is challenging in many situations yet the combination of machine learning with IMUs has shown promising results [11,32,34,35].

An example of a sport that proves promising for the application of IMUs and machine learning is American football. Currently, performance measures in American football are based on metrics that summarize a large amount of detailed data, sometimes for specific aspects of the sport. Over the years, these statistics became more advanced and have had an increasing impact on the match [36]. On the level of individual players, available metrics focus either on the overall performance of the player or on a specific performance aspect. One example of an overall metric is the quarterback rating, also called the passer rating. It summarizes the performance of a quarterback within a season. This metric is also referenced in the official NFL record and fact book every year [37]. In contrast to the NFL, the NCAA uses a slightly different algorithm to evaluate the passer rating [38]. Wide receivers are often evaluated with the defense-adjusted yards above replacement or the defense-adjusted value over average metric [39]. When considering the entire team, metrics focus on the team performance and sometimes more specifically on a subgroup of the team. Naturally, the offense team needs other metrics than the defence or special team. Also, within each part of the team, further distinctions may be useful. For example, a defence can be evaluated in their ability to stop runs of the opposing team early, but also the pass protection is of interest for an analysis. Metrics such as the sack rate, adjusted line yards, or second level yards, are typical for the defense line and linebackers [39].

Apart from metrics based on match statistics, players are regularly subjected to test scenarios in order to assess their performance. The “NFL combine” is an annual event in which selected athletes perform specific drills in order to establish an objective comparison. Tests such as the Wonderlic test [40], 20 and 60 yd shuttle, bench press, 40 yd dash, vertical and broad jump, etc. [41] deliver relevant data for scouts and coaches in the NFL. Based on this data, decisions for the draft are made, which is also reflected in [42]. The 40 yd dash especially is a predictor for the rank of athletes across all positions. Nevertheless, good results in the NFL combine do not correlate well with the performance of quarterbacks and wide receivers in the NFL [43].

All mentioned metrics are calculated from statistics of various actions and outcomes in the match or the NFL combine. Actions can be sacks, fumbles, drops, catches, pass attempts, etc. Outcomes can be a new first down, touchdown, turnovers, field goal, safety, etc. One drawback of the commonly used metrics is that they either rely on match data or the NFL combine. This excludes all athletes that are not part of a league that keeps track of all the relevant data. Additionally, all athletes that have low match time do not contribute significantly to the statistics and therefore the metrics for them are also not statistically significant. The lack of data in American football is problematic in some parts of the sport. On an amateur level, data is either not collected at all or only collected in an insufficient amount. Therefore, it is not possible to calculate meaningful metrics. To overcome this difficulty, one would also need to gather data during training to cover more athletes and obtain more detailed information about an athlete’s skill set. Additionally, new metrics for this new kind of data are necessary.

American football training is multifaceted in its exercises. Offensive plays in an American football match can be executed in various ways. A very common one is to pass the ball to a receiving player, catch the pass, and the run towards the end zone to score and win the match. All parts of this play are represented in training. Since there is a variety of exercises in training, necessary data for a potential neural network processing can vary for each respective exercise. One of the exercises in American football is about catching a pass, representing a key component in the match.

Catching the pass is a crucial point in a play and can decide whether a match is won or lost. Hence, choosing the right athlete as a receiver with the highest catch probability for a given play is very important. The decision process should be based, at least partly, on data. Therefore, data about catching, that contribute to a better decision, would be of interest. Unfortunately, the commonly used metrics and data on catching do not provide a good basis.

To calculate relevant information for coaches, scouts, etc. it is necessary to provide data on catching first. The fundamental measurement is about a classification of a catch or drop of an individual receiver. Based on that measurement, things like an individual catch probability or a receiver rating for various scenarios can be calculated. At first though, an adequate sensor platform has to be chosen and the corresponding algorithms have to be implemented.

Therefore, the main goal of this paper is to provide a sensor with a corresponding algorithm that allows receivers to automatically detect whether they caught or dropped a pass. The detection has to be reasonably reliable, which leads to the goal that at least 90% of all passes should be classified correctly overall. Since there is no need to classify the data in real time, no time limitations for the algorithm are given.

## 2. Materials and Methods

### 2.1. Approach to Measurement

To be able to classify a catch or a drop a sensor platform is needed that provides potentially important data. In general, there are several approaches to measure the relevant data. One possible way is to use a camera system that records the catching motion. With the video as input to an algorithm, it may be possible to classify a catch or a drop within the recorded video. A second possible approach is to use IMUs in wearables to monitor the movement of the arms of the receiver. With the motion data of the arms and a corresponding signal processing algorithm, a catch or a drop may be detectable within a sequence. A third possible approach is to record the sound during the catching process. Again, a signal processing algorithm would do the classification. Further options would be a mixture of the sensor (more data might lead to more reliable results), or an IMU within the ball that is caught.

The decision for this paper was to go with a wearable that provides several sensors, such as a 9 degrees of freedom (DOF) IMU, a pressure sensor and a microphone. Additionally, the platform had to be programmable so adjusting sampling rate, sensor sensitivity, synchronisation, data labeling, data storage etc. could be performed on the wearable, before or during the experiment.

### 2.2. Sensor Platform

The evaluation kit STEVAL-STLKT01V1 from ST Microelectronics was used as the sensor platform. The sensors on the platform are listed in Table 1 or in [44,45,46,47,48,49,50,51,52,53].

Further adaptions to the hardware were necessary to fit the experimental needs. Since each wrist of a pass receiving athlete wears one wearable, some sort of synchronisation algorithm had to be developed. Figure 1 shows the prototype of each version, with a more detailed setup of the prototypes in Figure 2.

As shown in Figure 1, two versions of the wearable have been developed. The first version, which uses wires and a pulse signal to trigger both wearables and synchronise their time stamps, has been developed earlier. The second version, which is based on wireless synchronisation and triggering of the wearables, provides more freedom for the athlete. Both versions were used in the experiment.

Each catch or drop delivers a data sequence, which needs to be labeled according to the outcome of the respective try to catch the pass. This labeling process was done differently for the two versions of the wearable. The wired version, as can be seen in Figure 1a, has one button on each wearable which is either pushed to label a catch or a drop. The devices were programmed in pairs, so one device was always configured with a catch label button, and the other one with a drop label button. According to Figure 1 additional colored tape indicated the respective meaning of the button to minimize mislabeling. The wireless version sends the data sequence to a mobile device, where it is labeled manually within an application, running on the mobile device.

### 2.3. Design of Experiment

The design goal of the experiment was to gather data on a broad spectrum of catches with a multitude of athletes and a wide range of possible conditions. In this case, a neural network, used as the signal processing algorithm, is forced to train its parameters on general signal information of catches and drops, not on specific ones from an athlete, pass, catching motion, or alike. For that reason, nearly no restrictions were given to the participants of the experiment within the task of data acquisition. The only relevant restriction was to try to catch the pass and do not fake drops, since the data should represent natural circumstances. Though, uncatchable passes, e.g., passes which are thrown not near enough to the receiver, are not considered a valid try, and should not be logged.

The experiment was reviewed and approved by the ethics board of the MCI. All participants were informed about the goal of the study, its risks and how the participants data will be processed. All participants signed a form of consent, which can be accessed by request.

To meet the design goal of the experiment, eight participant have carried out tests under various conditions. All of them were male with an age between 24 and 35. Their abilities to catch a pass are classified due to their further experience within sports and American Fooball. A participant, who has no ball sports related experiance, is classified as a novice. If the participant is familiar with sports such as basketball, baseball, handball or other ball-related sports, he is classified as advanced, since he has experience in estimating a trajectory of a pass [54,55,56,57,58] and is used to catch passes with other balls. A former or active player in American Football is classified as an expert.

Table 2 shows all relevant conditions that have been varied within the experiment. It underlines the multitude of conditions and athletes skills that contribute to a multifaceted set of data for training a neural network.

### 2.4. Data Acquisition

Prior to data acquisition, the wearables were fixed to the wrists. As can be seen in Figure 1, the wearables have labels with the text “left” and “right” on them. All participants were instructed to wear the wearables in an orientation so they could read the labels. This ensured that the wearables were not worn in a flipped manner, which would have led to rotated coordinate systems for some of the sensors.

As a first step in the data acquisition process, the sensor platforms had to be configured correctly in the initialization state. This was implemented trough a start up routine in the firmware of the sensor platform, which makes sure that every time the system is powered on, the system is configured in the same way. To monitor the motion during a catch or drop, the sensor platform was configured to meet the given sampling rates, range and resolution according to Table 1.

As a second state, the wearables are waiting for a trigger to record the whole data sequence. In this state, all signals were sampled continuously on both wearables and intermediately saved in a first in first out buffer on the random access memory (RAM) of the sensor platform. The buffer was configured to save approximately one second of data. If a trigger event happens, which reflects an absolute acceleration over 14 g, additional two seconds of all signals are sampled and recorded on the RAM. Both wearables were configured in a way such that if one wearable is triggered, also the other one is triggered. This is important to have synchronous data sets. One try to catch a pass leads to three seconds of data, one before and two after the trigger.

Depending on the version of the prototype, the labeling process, which follows the second state, was performed via the wearables or a mobile device. With prototype version 1 labeling was the next step in the data acquisition process after the data sequence was recorded. Directly after the end of the second state, a five second time window allowed the receiver to label the logged data as catch or drop via the button on the wearables. As mentioned, a catch was labeled by pressing the button of the green wearable and a drop by pressing the button of the red one. After the raw signals were logged and labeled, data was transferred to a local SD card. Version 2 of the prototype sends the logged raw data directly to a smart device, for labelling.

In total, 759 data sequences were recorded, using the procedure described above. This resulted in a dataset with 541 catches and 218 drops, covering the range of conditions given in Table 2. In the past, a classic approach for analysis would have been the extraction of hand-crafted features from the recorded sequences (e.g., Fourier features, wavelets, etc.) and the use of a support vector machine (SVM) for classification.

Although hand-crafted features and SVMs are still used to classify sensor signals in sports [32], fully parametric solutions are also able to identify similar (and more complex) features from training data. These fully parametric ML systems have been shown to outperform the hand-crafted approaches, if enough training data is available [59]. Since the present study meets this requirement, we opted for a convolutional neural network (CNN) to detect catches.

### 2.5. Preparation of Data for the Development of a Convolutional Neural Network

After data acquisition, all 759 data sequences had to be prepared for the neural network. Due to different sampling times and distribution of the data over two SD cards for the prototype version 1, an algorithm was used to create a standardized data shape, which can be used by the network. The algorithm uses linear interpolation to upsample the raw data to 8 kHz. To remove irrelevant data and artefacts from the data, the first 0.7
s are cut from all signals. Due to the outfall of the pressure signals in several data sequences no pressure signal was taken into the final data set for the network. Additionally, the data of both wearables is merged to one sequence, resulting in a matrix with a dimension of 18,400 × 20 for each signal as input for the network.

### 2.6. Neural Network Architecture and Implementation

The network should be able to identify unique features in data sequences that classifies the data as a catch or a drop. Feature extraction is a very common problem in computer vision and signal processing. Convolutional neural networks (CNN) for feature extraction are commonly used due to their good performance [60,61,62,63,64]. For this reason this paper also uses a model that is based on convolutional layers. A sequential model with multiple convolutional, average, max pooling, dense, and dropout layers to extract relevant features is used. This sequential model is illustrated in Figure 3. The given data is split in five equally sized sub sets, which all contain the same percentage of catches and drops. Four subsets were used for training the network, and one subset served as a validation set. Ranger [65] was used as the optimizer, with a maximum of 1000 epochs. An early stopping configuration with a patience of 20 optimization cycles and a minimum improvement of the validation loss of 1 × 10${}^{-3}$ per cycle was used to reduce the training time and prevent overfitting. As a loss function, binary cross entropy was used since the networks output is binary (either a catch or a drop). As an activation function in all layers, except for the output layer, a parametric rectified linear unit (PReLu) [66] was used. The output layer uses a logistic sigmoid as its activation function. In total, the network has 2,198,365 trainable parameters.

## 3. Results and Discussion

To get a better understanding of the catching motion, the first findings in the signals have not been identified by computer aided analysis, but by observation of an average try to catch a pass and the corresponding signals. Among the participants of the experiment, several approaches to catch a pass were observed. The novice participants lacked a good catching technique, which resulted in more drops on average in comparison to the expert and advanced level athletes. Nevertheless, the process of catching was the same for all athletes. The catching motion starts with a unique starting pose for each athlete. After the pass is released the athlete starts to position himself according to his estimation of the ball’s trajectory. In this positioning phase, the novices tended to have a sharper movement, due to misjudgment and readjusting, resulting in higher accelerations, in comparison to the advanced and expert athletes. When the ball establishes contact with the hands for the first time, the catching phase starts. This can be observed in the data sequences as strong amplitudes in acceleration, spin rate, volume, etc. A few hundred ms after the impact, the catching phase ends and the post impact phase starts. During this phase, a variety of different signal shapes occur. The shape highly depends on the scenario, athlete and the outcome of the catching process. Assuming the cause-effect relationship, especially the catching and post impact phase contains information about a catch/drop. Although it was possible to characterise the phases of the catching process, it was not possible to find clear and easily visible indicators for a catch/drop within the signals, with classic methods of signal processing such as thresholding, Fourier and wavelet analysis. This finding underlines the need for neural networks, since the classification is not easily possible with conventional signal processing.

### 3.1. Raw Data and Input to the Neural Network

The collected dataset is too large to visualize it completely, therefore only an excerpt is given in Figure 4. This excerpt also is an example for an illustration of the input to the neural network. Note, that input data highly depends on the conditions listed in Table 2. Figure 4a shows the left hand and Figure 4b the right hand signals, which are acceleration *a* in all axis, spin rate ω around all axis, flux density *B* in all axis and quantized audio. Feel free to contact the authors for access to the whole dataset.

### 3.2. Neural Network Performance

To train the neural network, the dataset was split into five nearly equally sized folds with a stratified approach. Four folds have been used for training the network, one fold was used as a test fold. Fifty training iterations with four trainings in each iteration for each of the training folds have lead to an average classification accuracy of 93% on the test fold of each iteration.

This accuracy is also reflected in the confusion matrices, as can be seen in Figure 5. Since the data of Figure 5a contains also the folds, which the network used for training, the accuracy exceeds 93%.

During the training iterations, the mispredicted data sequences were logged. This has been done to identify mislabeled or badly triggered data sequences, but also to find data sequences that are difficult to classify. During fifty iterations, several observation were made. Figure 6 shows the distribution of the mispredicted data sequences over their reccurance, i.e., the amount of times a single data sequence is classified incorrectly during the verification stage of each 50 training iterations. As can be seen in Figure 6 eight data sequences have been mispredicted repeatedly during the fifty iterations and most of the mispredicts occurred less than ten out of fifty times.

As a limitation of the system, the triggering has to be considered. The system triggers with an absolute acceleration of 14g. This leads to the problem that a catch attempt might not trigger the sensor platform, due to not reaching the threshold of 14g. This can happen e.g., if no appropriate contact was made with the passed ball during a catch attempt. A possible scenario would be a body catch, where most of the kinetic energy of the pass is taken by the body, but not the hands or wrists. Since this scenario may not trigger the sensor platform, it cannot be classified. Another scenario would be if the pass never makes contact with the hands or wrists of the receiver, although a catch attempt was made. Clearly this should be classified as a drop, but since the triggering needs a threshold of 14g, it will not be recorded and therefore also not classified. In general, the mentioned scenarios may happen more often to novice players, who lack a good catch technique.

Another limitation for the use of such sensor would be mistriggering. There are possible scenarios in which triggering can happen due to other impacts. This might occur, if athletes clap their hands, or an opposing player tackles them. All these scenarios would be falsely classified as a drop or a catch. Therefore the system should be applied in training but not in a match setting to maintain high classification accuracy. Additionally this limit might be pushed if the network is extended to have three outputs (catch, drop, no attempt), instead of only two. For this study, this was not possible due to no adequate data.

### 3.3. Significance of Inputs

Further analysis of the dataset and the significance of the sensor data is of interest, due to a possible optimization of the wearable and the neural network. For that reason an analysis was carried out and the results are shown in Table 3. Note, that for this analysis, five iterations were used in each training to identify the mean accuracy.

As can be seen in Table 3 the magnetometer sensor and the acceleration sensor seems to be of highest significance. The least important sensor data are the audio signals. This can be used to further simplify the network, since especially the audio signals have to be sampled with the highest sampling rate. Therefore, neglecting the audio signals decreases the inputs order and size, which also has the potential to decrease the number of trainable parameters in the neural network considerably.

The sensor board, that was used in this study, also has limitations concerning the sensor signals. Since the acceleration and magnetometer sensor have the highest significance, the limitations of the range of the signals limits the ability for classification slightly. In Figure 4 saturation of the acceleration is visible at approximately 0.4
s. The acceleration is limited due to the sensor limit of ± 156.96
m/s. With the introduction of an acceleration sensor with a wider range additional information may be gathered that could lead to a better accuracy. Nevertheless, the given data set already shows results, that satisfy the original goal of at least 90% correctly classified passes.

## 4. Conclusions

The main goal of this paper, i.e., to develop a sensor platform in combination with a neural network to classify catches and drops of receiving players in American football accurately, has been met. Additionally, the paper provides a new type of dataset, which is unique in its composition and creation.

From a sports science perspective, new possibilities open up because of the findings of this paper. A sensor for catch detection, in combination with an automated passing machine such as the one developed in [31], enables fully automated catch training, which allows for various new training possibilities. For example, an automated passing machine could aim on a designated spot where a receiver should attempt to catch a pass. By automatically measuring the success of the catch attempt statistics can be gathered, which link specific catch scenarios (one hand catching, over the shoulder, left and right, etc.) with individual catch success rates. Therefore, it is possible to determine strengths and weaknesses of an athlete for specific catch scenarios. This is important information for any coach, since the exercises can be optimized for all athletes individually.

Information about the catch rate in training in various scenarios might serve as the decision basis of a coach, which receiver should run which route in a play. This way, it is possible to maximize the success rate of a pass play.

The sensor additionally enables continuous monitoring of catch attempts during training, which enables analysis of potential and progress in catching abilities. From a scouts perspective, this might be relevant information about an athlete.

## Figures and Tables

**Figure 1 sensors-20-06722-f001:**
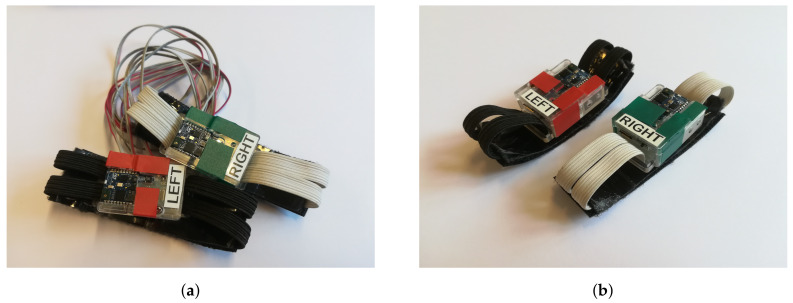
Prototype Version 1 uses general purpose inputs and outputs for synchronisation and labeling, hence a button for each board and a wire to connect the wearable on each wrist were necessary. For version 2 there was no further need for this. (**a**) Prototype Version 1, (**b**) Prototype Version 2.

**Figure 2 sensors-20-06722-f002:**
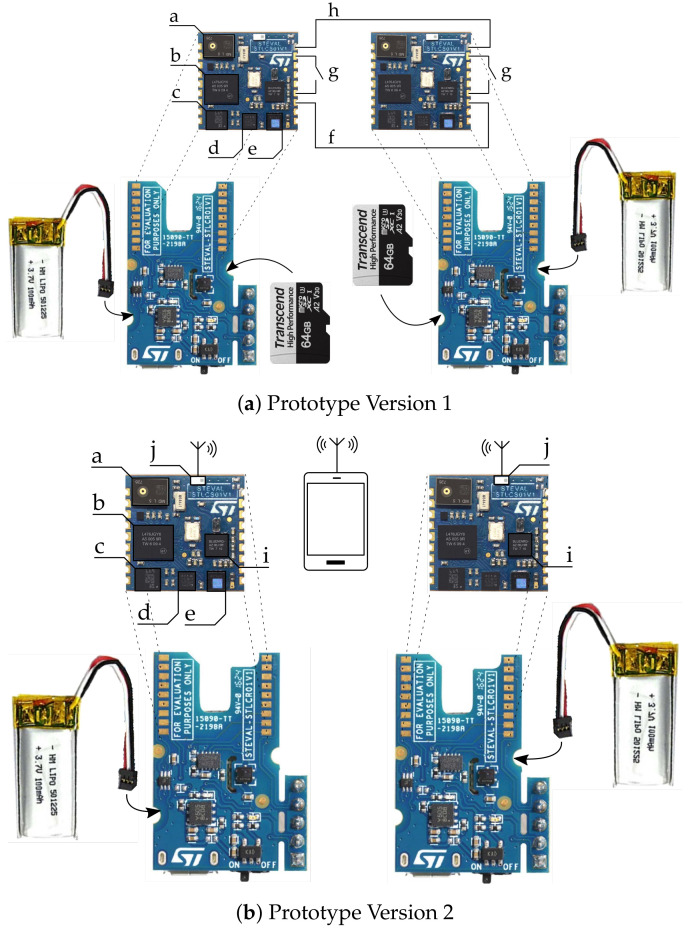
Two prototypes were developed for the experiment. The components on the printed circuit board (PCB) are indexed as follows with a: microphone, b: microcontroller, c: accelerometer and gyroscope, d: magnetometer, e: pressure sensor. (**a**) shows the first version of the wearable, with a wired synchronization and data labeling setup as the components are indexed with f: synchronization wire, g: labeling button, h: common ground. (**b**) shows the setup with a wireless synchronization via Bluetooth low energy (BLE) with the components indexed as i: BLE Communication chip and j: Antenna. The labeling of the data is done in an application running on the mobile device.

**Figure 3 sensors-20-06722-f003:**
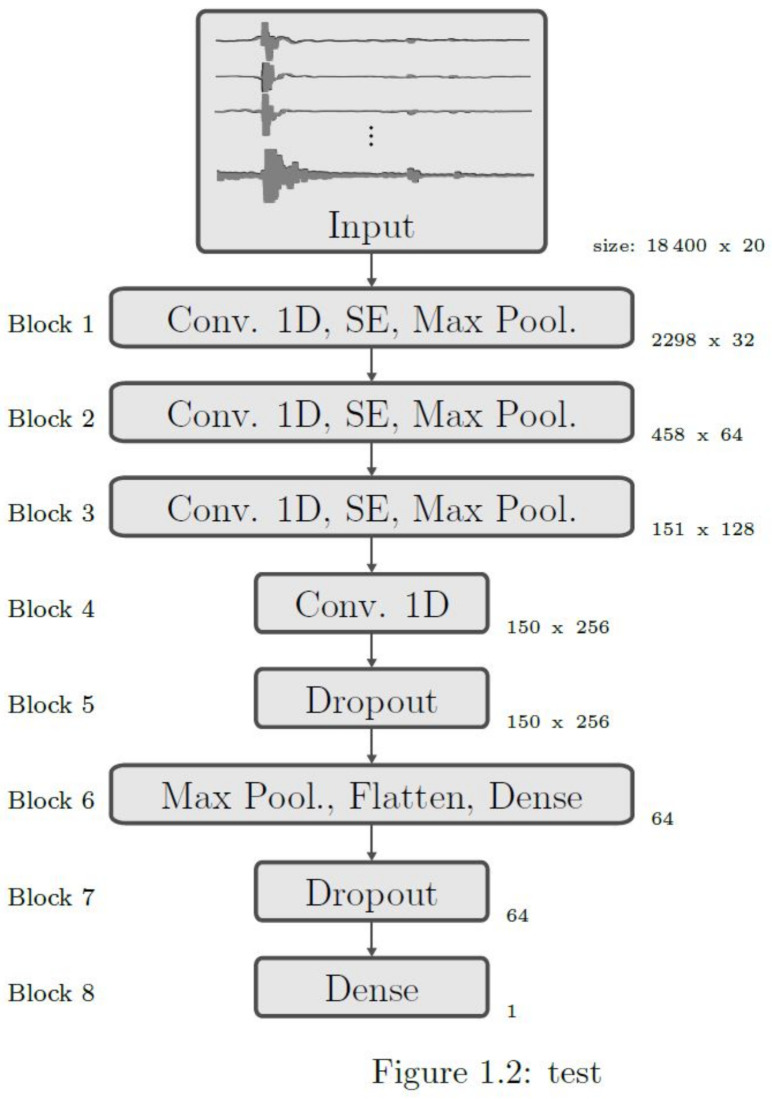
The neural network architecture used for the experiments in this paper. Blocks 1 to 3 consist of a convolution layer with additional layers such as squeeze and excitation layer and max pooling layer.

**Figure 4 sensors-20-06722-f004:**
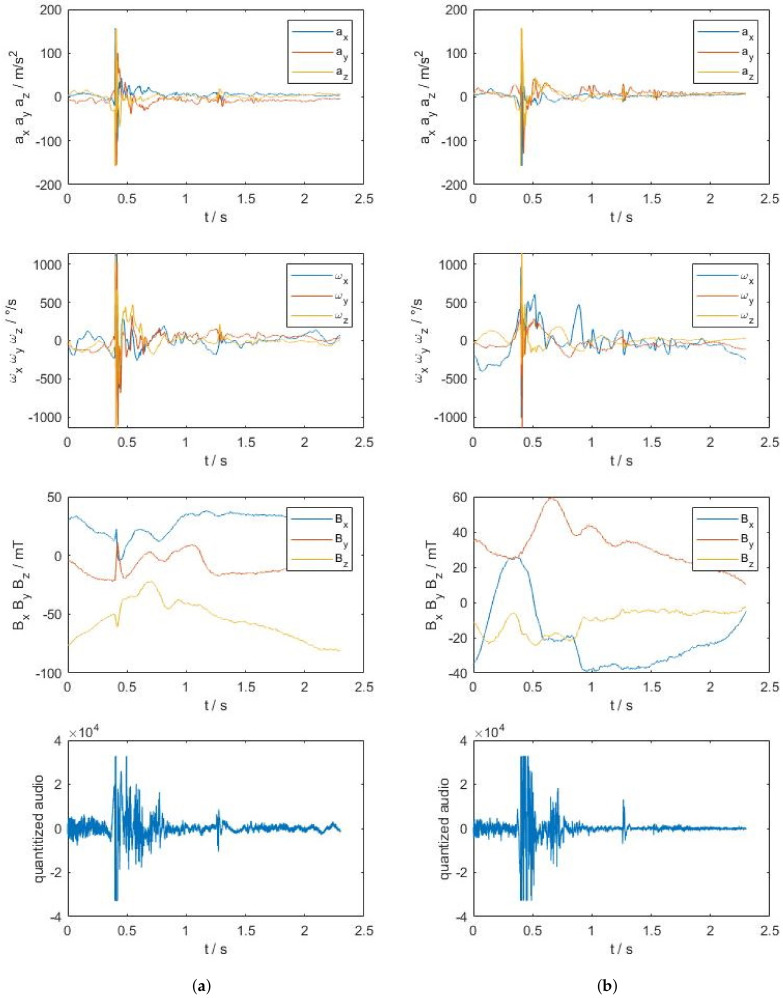
Raw signals (acceleration a in all axis, spin rate ω around all axis, flux density B in all axis and quantized audio) of the left (**a**) and right (**b**) hand for a caught pass under the circumstances: automated passing machine [31], standing, facing towards, twohanded, advanced, indoor, Prototype 1.

**Figure 5 sensors-20-06722-f005:**
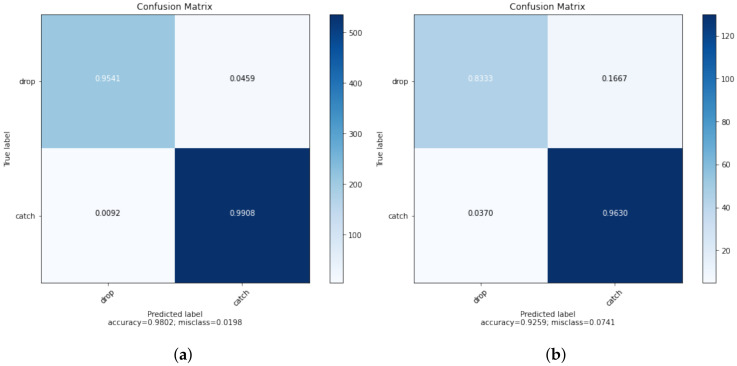
The normalized confusion matrices show the classification accuracy as comparision between the predicted catches/drops and the real catches/drops. (**a**) shows the matrix of the whole dataset, whereas (**b**) shows the matrix of the validation fold in the last iteration.

**Figure 6 sensors-20-06722-f006:**
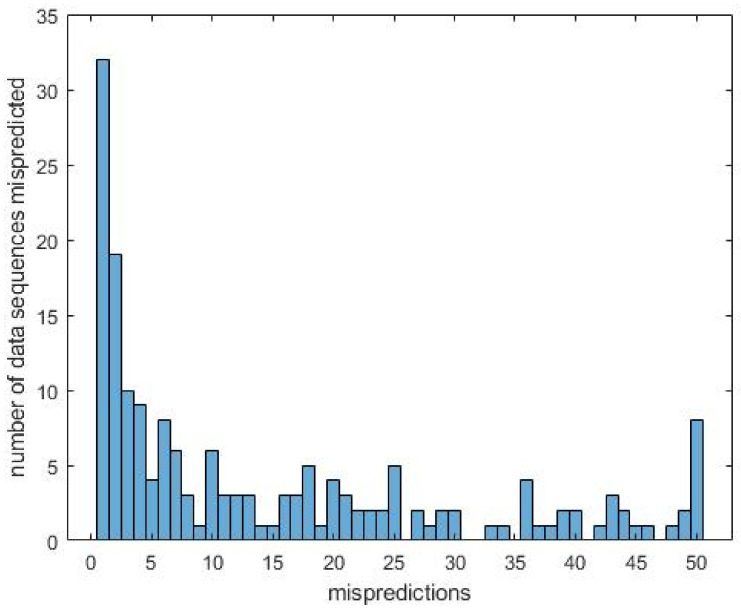
This histogram shows the 2630 mispredicted data sequences distributed over the accumulated mispredictions of the data sequences.

**Table 1 sensors-20-06722-t001:** Settings of the Sensors on the STEVAL-STLKT01V1 for the experiment.

Sensor Code	Sampling Frequency $f_{s}$	Range	Resolution	Usage
LSM6DSM	1 kHz	±16 g	16 bit	acceleration
LSM6DSM	1 kHz	±1000 ${}^{\circ }$/s	16 bit	spin rate
LSM303AGR	100 Hz	±5 mT	16 bit	magnetometer
LPS22HB	100 Hz	260–1260 hPa	16 bit	absolute pressure
MP34DT05-A	8 kHz	0–122.5 dbSPL	16 bit	audio

**Table 2 sensors-20-06722-t002:** Various conditions for the experiment.

Condition	Range
pass source	human quarterback, automated passing machine [31]
movement scenario	standing, running left/right/towards/away from the passer, jumping
positioning scenario	facing towards, sideways, away from the passer
pass scenario	passes thrown high and low, off target, over the shoulder
catch scenario	one/two handed, body catch (first impact is on the body)
athlete level	four novice/three advanced/one expert
external	brightness levels from approximately 50–80,000 lx, indoor and outdoor tests, experiments with and without wires running from one wearable to the other via the sleeves (due to the different prototype versions)

**Table 3 sensors-20-06722-t003:** Mean accuracy of the neural network with various input settings with five iterations.

Training	Accelerometer	Audio	Gyroscope	Magnetometer	Accuracy
1	on	on	on	on	92.99%
2	zeros	on	on	on	91.09%
3	on	zero	on	on	92.86%
4	on	on	zero	on	93.15%
5	on	on	on	zero	92.33%
6	on	zero	zero	zero	91.91%
7	zero	on	zero	zero	75.44%
8	zero	zero	on	zero	86.80%
9	zero	zero	zero	on	91.30%
10	on	zero	zero	on	93.39%

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
