# Peer review of "Using Wearable Sensors and a Convolutional Neural Network for Catch Detection in American Football"

_sensors, 2020, doi:10.3390/s20236722_

Round 1
Reviewer 1 Report
The authors present interesting and very concrete work to improve the collection and integration of data in American football. Today, the integration of data that reflects physical, technical and tactical performance in team sports is absolutely essential to analyze the game and achieve greater performance and efficiency. The work could be improved to provide greater utility and implications in this area: performance monitoring, big data and sport.
Major changes:
The authors do not carry out an analysis of the context and area of action in accordance with the current reality and latest innovations in the field of performance monitoring and data use. The authors mix technologies for completely different uses and from different areas of sports analysis. You cannot mix systems such as the electrocardiogram or force plates with video analysis. They are different things from different areas. Therefore, technologies cannot be named randomly to justify the work carried out.
From the point of view of the reviewer, the authors work in the field of technology for performance analysis first, and within that area, in the area of EPTS (Electronical Performance and Tracking Systems). It is not possible to deduce this at any point in the article. See https://digi-sporting.eu/wp-content/uploads/2020/07/Handbook.pdf
Therefore, it is recommended that the authors improve the wording of the introduction following a clearer common thread. First, talk about the importance of technology for performance monitoring in sport, as well as the usefulness of performance monitoring itself. Second, a small analysis of one of the most developed areas should be deepened: EPTS. EPTS are mainly composed of Optical-based tracking system, Local Positioning System (LPS) and GPS / GNSS satellite system. In the case of LPS, GPS and GNSS, other technologies are also incorporated that provide additional information such as accelerometers, heart rate monitors or gyroscopes. The third place, once the context has been addressed, we can talk about the new utilities of IMUs, the object of the article.
Finally, it is absolutely necessary to talk about the technical action to be analyzed (catch or drop) within the sport, and what use it can bring to improve performance.
Minor changes:
Line 24. The videos of public broadcaster are not a reliable and good resource to obtain performance data nowadays. The authors should talk about the “Optical-based tracking system”, typical of the major sports leagues and that provide powerful analysis software for the coaching staff of the teams.
Line 25. IMUs have nothing to do with video analysis, as mentioned before.
Is better to use mathes instead games.
If the authors consider it, I think it would be very convenient to include a last section or paragraph on how this idea or prototype could be implemented in American football or other sports, on a commercial level.
Author Response
Dear reviewer,
Thank you for your valued feedback and the possibility to improve our manuscript accordingly. Attached to this message you can find our response to your review. In most points we followed your review and tried to include your feedback in our revised manuscript. We are looking forward hearing your review on the revision.
BR Bernhard Hollaus

Reviewer 2 Report
The paper titled “Using Wearable Sensors and a convolutional neural network for Catch Detection in American Football” presents a sensor platform based on a 9-dof IMU, acoustic sensor, and pressure sensor to classify catches and drops in American football passes. The work is novel and overall the methodology is at a high level sound, though I have some questions / concerns with the work that I have outlined in my major comments. There are also a few minor points that are mostly clarification.
Major Comments:
- The introduction I think can be cleaned up a little bit. The discussion of cameras and force plates don’t seem terribly relevant to the paper, as particularly for the force plates, these are not being utilized in the present work. I would have preferred the authors jump straight into discussing what measurements are made specifically in American football since that is the sport that is the focus of this paper, and use this to motivate the lack of measurements in catching as an important skill to track. Here is what I would recommend, though I won’t be offended if the authors choose not to follow this exact outline:
- First paragraph is fine, discussing the importance of tracking performance in sports
- I would then go straight into American football and talk about what performance metrics are most useful
- Then I would discuss papers that track these performance metrics and what sensors they use (running speed / form is naturally important given that it is tested in the NFL Combine for example. Head impacts are also important to track given the recent attention on concussions, etc.)
- Then I would talk about catching as a skill that is not currently tracked automatically and still has to be done manually to motivate this particular study.
- If the authors do choose to keep the format of their introduction largely intact, I would like to see some justification. If the authors do choose to keep the introduction intact, there are many citations missing in the introduction when the authors are discussing past sensors and how they are applied in various fields. I’ve listed the ones I think are necessary here, but the authors would do well to back up any specific claims with citations.
- Page 1, lines 30-32: Citations for high speed cameras
- Page 2, line 35: Citation for depth cameras? Kinect has traditionally been the most popular in this category so there should be some citations here.
- Page 2, line 39: Add citation or these IMU placements. Placement of IMU is also common directly to the skin, as is done in citation 23 and similar works in the head impact measurement space.
- Page 2, line 40-43: The authors need citation for the sampling rate and high accuracy arguments in cricket and gait respectively. These seem like very specific examples so I assume they do come from some prior work / knowledge.
- Page 2, line 43: What are flux densities? I assume the authors are referring specifically to the magnetic flux densities that indicate heading? Note, the magnetic flux is only useful for determining heading direction, but orientation with respect to gravity is done with the accelerometer typically. There are papers that deal with measuring orientations without a magnetometer, albeit with less accuracy ([1] as an example)
- Page 2, line 44: Authors need citations for these filters, ideally in the context of IMUs.
- Page 2, lines 45-46: Citations for force plates in these applications
- Page 2, lines 48: Citations for vertical force in jumping
- Page 2, line 49: Citation for all forces required for locomotion
- Page 2, line 50-51: Citation needed for differentiating sampling rate in different activities
- Page 2, line 52: citation needed for force plate array in sprinting
- Page 2, line 55: citation using neural network in processing sport data
- Materials and methods: The authors begin by stating different sensor options for detecting catches, but imply that all of these sensors must be paired with neural networks to perform detection. Why do neural networks absolutely need to classify this data? I don’t see any reason that a neural network needs to be the algorithm underlying classification algorithms. There currently are several works in the head impact space that use similar IMU pre-processing steps (i.e. thresholding the IMU signals to identify an event) and non-neural network algorithms to identify head impacts [2]-[3]. The authors should better justify their use of a neural network here.
- Note, I just read in the results that this is briefly hinted at in page 7, lines 209-210. Still, I would be careful in defining what “conventional signal processing” refers to in this case as it is implied that the authors did not attempt to classify using “conventional signal processing” techniques but merely by visibly looking at the kinematic traces.
- Ethics approval for human subjects: As this study clearly collected data from human participants, the authors need to state the ethical board that approved this study and specifically whether informed consent was obtained from the participants.
- Similar to this, while the authors do a good job stating the prior experience with catching for participants, perhaps the age range and gender of participants would also be useful.
- Sensors: Were the sensors always placed in the same way on the hand? IMUs in particular measure data in their own reference frame, so if you have a sensor with the “right” and “left” text labels close to the wrist for one participant but closer to the knuckles on a different participant, then the kinematic signals will look quite different. If the sensors are placed always in the same orientation on the hand for all participants, then will the system easily generalize to consumers who may not be so diligent in placing sensors correctly?
- Ground truth labels: For the button version, who is pushing the button to label a catch or a drop? Is it the player? How reliable is this method? For the wireless version, who is making the assessment of catch or drop? Presumably the person using this app can actually see what is happening and observe whether a ball is caught or not? It is common in the NFL to have replays of questionable catches, such as when a player dives. How confident are these catch labels in those scenarios?
- Page 6, 166-167: Are the authors concerned that the dataset is unbalanced (i.e. more catches than drops)? Unbalance datasets can sometimes introduce bias (e.g. if there are 90% catches and 10% drops then classifiers can just declare everything a catch and have a 90% true positive rate). Admittedly the unbalance in this dataset is not egregious, though the confusion matrices suggest that the algorithm is biased towards labeling catches.
- Page 6, line 173: What is the reasoning for not including the pressure data? This seems almost arbitrary. I would have liked to see a more thorough analysis to argue for not including the pressure data, similar to what you did with all of the other sensors in table 3.
- Figure 4: According to the axes on your plots and the specs presented in table 1, your data signals for acceleration, spin rate, and audio are clearly clipping right around the 0.4-0.5s mark (e.g. the signals are reaching the measurement limits of your sensors). Is this common in your data? I imagine this could be a problem as you are losing the ability to measure the true peak movements. Whether this will affect your classification is unclear.
- Also now that I think about it, the 14g threshold is pretty much right at the limits of the accelerometer. I assume this is a threshold in any of the x, y, or z acceleration signals as opposed to an acceleration magnitude threshold, given that the authors use a positive / negative threshold.
- This data augmentation bit is sort of tossed in the results and not described in great detail, and as far as I can tell the results aren’t even presented. If the authors want to include data augmentation, they should describe thoroughly the parameters of augmentation (what is the magnitude or random noise, what are the range of time shifts, what is the range of magnitude scaling parameters), how much augmented data was added, and the actual results following data augmentation to compare against un-augmented data. As it is written, the data augmentation paragraph serves no real purpose and detracts focus away from the main results that are presented in detail.
- I would have liked to see a discussion section describing the implications of the results, potential future use of this technology, and any limitations the authors might have.
- One major limitation I would like discussed is the 14g threshold to identify a catch attempt. How accurate is this at detecting actual attempts and also not capturing non-catch attempts? Thresholding on linear acceleration is a common way to identify similar signals, such as steps in step counting devices and head impacts in the concussion field [2]-[3]. However, these thresholding signals are notorious for being very sensitive to spurious unrelated movements, resulting in false positives, and if the threshold is tuned improperly, could also result in undetected events (false negatives). As I’m more familiar with the head impact space, there have been some work characterizing these with videos [4]-[6]. The authors present a solid case for identifying whether an identified catch attempt was successful or not, but I would be very careful claiming this tool as a catch detector, as there is no evidence suggesting that the initial identification of a catch attempt is accurate or not. Thus, for example, a player attempts a one-handed catch and does not ever make contact with the ball. In the absence of any contacts resulting in high accelerations, the sensors would not exceed threshold and thus will not trigger, but a pass attempt was made.
Minor Comments:
- Page 2, line 54: Can the authors be more detailed on these “specific algorithms?” Without looking up the referenced citations and reading them it is not clear what is common in data processing here.
- Page 3, line 115-116: Wireless and wired triggering are inherently different, with the wireless time synchronization being particularly tricky, especially for fast signals like the 1000Hz IMU and 18kHz audio. Did the authors attempt to validate this synchronization? The easiest experiment I can think of is to place both wrist sensor IMUs on a rigid object and spin it around. The measured gyro signals should match precisely if there is no lag according to rigid body dynamics. If there is a lag, it should be easy to quantify using this method and a cross-correlation analysis (xcorr in matlab).
- Page 7, lines 196-197: Are there numbers for average catches and drops among the groups? This seems relevant as perhaps the drop data is biased towards novices and the catch data is biased towards experts.
[1] Won SH, Melek WW, Golnaraghi F. A Kalman/particle filter-based position and orientation estimation method using a position sensor/inertial measurement unit hybrid system. IEEE Transactions on Industrial Electronics. 2009 Sep 22;57(5):1787-98.
[2] Wu LC, Kuo C, Loza J, Kurt M, Laksari K, Yanez LZ, Senif D, Anderson SC, Miller LE, Urban JE, Stitzel JD. Detection of American football head impacts using biomechanical features and support vector machine classification. Scientific reports. 2017 Dec 21;8(1):1-4.
[3] Wu LC, Zarnescu L, Nangia V, Cam B, Camarillo DB. A head impact detection system using SVM classification and proximity sensing in an instrumented mouthguard. IEEE Transactions on Biomedical Engineering. 2014 Apr 25;61(11):2659-68.
[4] Campbell KR, Marshall SW, Luck JF, Pinton GF, Stitzel JD, Boone JS, Guskiewicz KM, Mihalik JP. Head Impact Telemetry System’s Video-based Impact Detection and Location Accuracy.
[5] Patton DA, Huber CM, McDonald CC, Margulies SS, Master CL, Arbogast KB. Video confirmation of head impact sensor data from high school soccer players. The American Journal of Sports Medicine. 2020 Apr;48(5):1246-53.
[6] Miller LE, Pinkerton EK, Fabian KC, Wu LC, Espeland MA, Lamond LC, Miles CM, Camarillo DB, Stitzel JD, Urban JE. Characterizing head impact exposure in youth female soccer with a custom-instrumented mouthpiece. Research in Sports Medicine. 2020 Jan 2;28(1):55-71.
Author Response

(The authors gave the same response as above.)

Round 2
Reviewer 2 Report
I would like to thank the authors for addressing all of my comments from the last revision. What I have here are relatively minor comments.
- Page 7 of 15, lines 194-198. This is quite a strong statement regarding the trends in machine learning. I would be careful making such bold statements. Indeed, systems such as neural networks that can learn the appropriate features can be more generalized; however there are other drawbacks to these systems as well. For one they are more difficult to develop and train and as the authors noted, require large amounts of data to avoid overfitting (the paper that is cited discusses training on millions of images, compared to the <1000 that are in the paper, though hard to compare since the architectures are different). I think the author’s justification for using neural networks is fair, though I would be more careful about making blanket statements that “SVMs have fallen out of fashion in the machine learning community” because simpler techniques (even linear regression) still have their place in classification problems that do not require the complexity of neural networks.
- Page 12 of 15, line 310-321. I like these broader use cases for the sensor that the authors introduced.
Author Response
Thank you for your valued feedback. Please see our response to your review in the attached file.
